# Dissection of central clock function in *Drosophila* through cell-specific CRISPR-mediated clock gene disruption

Rebecca Delventhal[1†‡], Reed M O'Connor[1†‡], Meghan M Pantalia[1†‡], Matthew Ulgherait[1], Han X Kim[1], Maylis K Basturk[1], Julie C Canman[2], Mimi Shirasu-Hiza[1]*

[1]Department of Genetics and Development, Columbia University Medical Center, New York, United States; [2]Department of Pathology and Cell Biology, Columbia University Medical Center, New York, United States

*For correspondence:
ms4095@columbia.edu

[†]These authors contributed equally to this work
[‡]The first three authors are listed in alphabetical order

Competing interests: The authors declare that no competing interests exist.

**Abstract** In *Drosophila*, ~150 neurons expressing molecular clock proteins regulate circadian behavior. Sixteen of these neurons secrete the neuropeptide Pdf and have been called 'master pacemakers' because they are essential for circadian rhythms. A subset of Pdf⁺ neurons (the morning oscillator) regulates morning activity and communicates with other non-Pdf⁺ neurons, including a subset called the evening oscillator. It has been assumed that the molecular clock in Pdf⁺ neurons is required for these functions. To test this, we developed and validated Gal4-UAS based CRISPR tools for cell-specific disruption of key molecular clock components, *period* and *timeless*. While loss of the molecular clock in both the morning and evening oscillators eliminates circadian locomotor activity, the molecular clock in either oscillator alone is sufficient to rescue circadian locomotor activity in the absence of the other. This suggests that clock neurons do not act in a hierarchy but as a distributed network to regulate circadian activity.
DOI: https://doi.org/10.7554/eLife.48308.001

## Introduction

Circadian rhythms are 24-hour oscillations in physiological functions and behaviors, including locomotor activity, immune system function, metabolism, and sleep (*Allen et al., 2016*; *Ulgherait et al., 2016*; *Stone et al., 2012*; *Hill et al., 2018*; *Shirasu-Hiza et al., 2007*; *Panda, 2016*; *Vaccaro et al., 2017*). Disruption in circadian regulation is a common feature of aging and is associated with a variety of adverse health outcomes such as diabetes and cancer (*Rosbash and Takahashi, 2002*; *Maury et al., 2010*; *Turek et al., 1995*; *Wulff et al., 2010*). Circadian rhythms are driven by 'molecular clocks,' or proteins that regulate rhythmic gene expression. Work in *Drosophila* has been crucial for understanding the molecular clock, a transcriptional negative feedback loop with four core proteins: Clock, Cycle, Period, and Timeless (*Figure 1A*) (*Allada et al., 1998*; *Hunter-Ensor et al., 1996*; *Rutila et al., 1998*; *Sehgal et al., 1994*; *Sehgal et al., 1995*; *Vosshall et al., 1994*). In brief, Clock and Cycle activate transcription of *period* and *timeless* which, once translated, dimerize and translocate into the nucleus where they bind to Clock and Cycle, thereby inhibiting their own transcription; this molecular feedback loop repeats with a 24-hour periodicity (*Figure 1A*). Importantly, the core components of the molecular clock in *Drosophila* are conserved in humans (*Ch and Takahashi, 2006*).

In *Drosophila*, ~150 neurons in the brain have molecular clocks and control circadian locomotor activity (*Figure 1E*) (*Top and Young, 2018*). These clock neurons cluster in eight subgroups defined by their anatomical locations: small and large ventral lateral neurons (s-LNvs and l-LNvs), the 5th s-LNv, dorsal lateral neurons (LNds), lateral posterior neurons (LPNs), and three separate clusters of

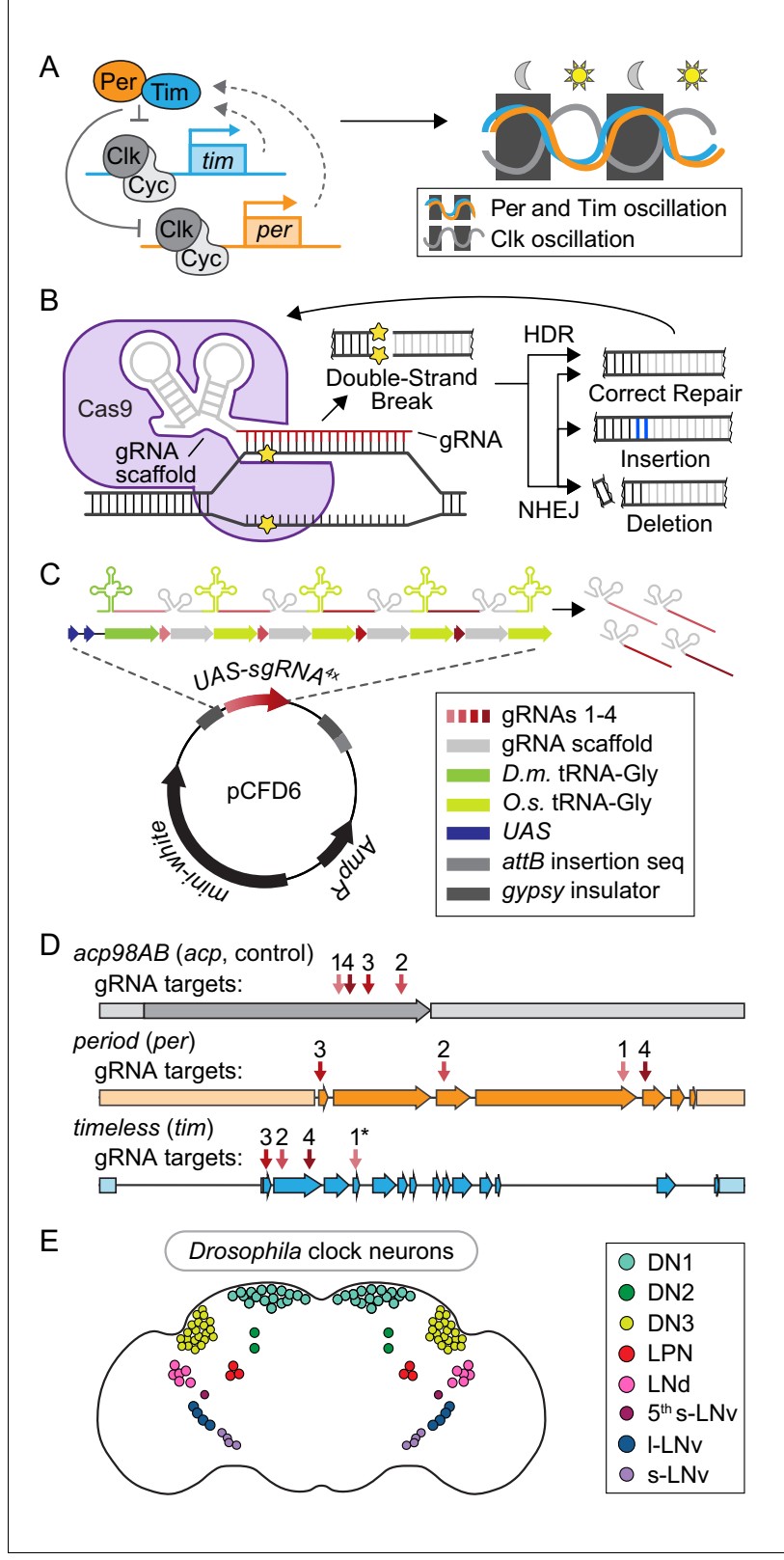

**Figure 1.** Toolbox for cell-specific, CRISPR-mediated disruption of core circadian regulators. (**A**) Schematic of the transcriptional/translational negative feedback loop that drives rhythmic expression and activity of the four core circadian regulators: Period (Per), Timeless (Tim), Clock (Clk), and Cycle (Cyc). (**B**) Diagram of CRISPR-Cas9 mediated DNA damage and repair pathways. (**C**) Diagram of plasmid (pCFD6, adapted from ***Port and Bullock,***

*Figure 1 continued on next page*

*Figure 1 continued*

**2016**) used to generate *UAS-sgRNA⁴ˣ* transgenic flies. *D.m.* = *Drosophila melanogaster*. *O.s.* = *Oryza sativa*, Asian rice. (**D**) Diagram showing sgRNA target sites for *acp98AB* (*acp*, gray), *period* (*per*, orange), and *timeless* (*tim*, blue), numbered in order of 5′−3′ position in the respective *UAS-sgRNA⁴ˣ* construct. Arrows = exons; shaded rectangles = promoters and UTRs. \**tim* sgRNA one has a single base pair deletion in the Cas9-binding scaffold region (see Materials and methods). (**E**) Diagram of ~150 clock neurons organized into the following anatomical and functional clusters in the *Drosophila* brain: dorsal neurons (DN1, DN2, DN3), lateral posterior neurons (LPN), dorsal lateral neurons (LNd), and small and large ventral lateral neurons (s-LNv, 5th s-LNv, l-LNv).
DOI: https://doi.org/10.7554/eLife.48308.002

dorsal neurons (DN1s, DN2s, and DN3s) (*Figure 1E*). Cell ablation and cell-specific rescue experiments identified two sets of clock neurons that control circadian locomotor activity: Pdf⁺ s-LNvs comprise the 'morning oscillator' and control the morning peak of activity, while the 5th s-LNv and LNds comprise the 'evening oscillator' and control the evening peak of activity (*Stoleru et al., 2004*; *Grima et al., 2004*; *Yao et al., 2016*). In the classic paradigm of circadian neuronal circuitry, the morning oscillator neurons are thought to be master regulatory neurons that synchronize molecular clocks in other neurons via rhythmic release of the neuropeptide Pigment-dispersing factor (Pdf) (*Top and Young, 2018*; *Renn et al., 1999*; *Helfrich-Förster, 1995*; *Fernández et al., 2008*; *Shafer et al., 2008*). However, a subset of *Pdf* mutants (~25%) were reported to retain rhythmic activity with a shortened period (*Grima et al., 2004*) and more recent experiments involving cell-specific expression of period-lengthening and shortening genes have suggested that circadian neurons interact through a complex network, rather than a hierarchy, to regulate circadian behavior (*Yao et al., 2016*; *Yao and Shafer, 2014*). The precise role of molecular clock components in these circadian-regulatory neurons remained unclear.

To assess the role of molecular clock components in specific clock neurons, researchers have typically used the Gal4-UAS system for cell-specific RNAi-knockdown of clock genes and cell-specific rescue in a null mutant (*Martinek and Young, 2000*; *Shafer and Taghert, 2009*). While instrumental in understanding neuronal control of circadian behaviors, these strategies have limitations. RNAi can be inefficient: Martinek and Young observed only ~50% reduction in *per* RNA levels with eye-specific RNAi knockdown of *per* (*Martinek and Young, 2000*). Moreover, unlike *per* null mutants, which are 100% arrhythmic, flies with *per* RNAi knockdown in all Tim⁺ cells were shown to be only 45% arrhythmic (*Ng et al., 2011*) or rhythmic with lengthened period (*Martinek and Young, 2000*). Similarly, cell-specific rescue experiments sometimes do not reproduce wild-type rhythmic behavior, possibly due to constitutive expression of normally rhythmic genes. Pan-neuronal or ubiquitous rescue of *per* or *tim* in a null mutant background caused variable rhythmicity (~50–95%), depending on the UAS transgene insertion and Gal4 driver lines used; even overexpression of *per* and *tim* in a wild-type background sometimes resulted in a partial loss of rhythmicity (*Yang and Sehgal, 2001*). Thus, while cell ablation experiments have shown the necessity of specific neurons for regulation of circadian locomotor activity, the function of the molecular clock within those neurons remains unclear.

Recent advances in CRISPR technology in *Drosophila* provided an opportunity to create new tools for circadian research (*Gratz et al., 2013*; *Yu et al., 2013*). One key advance was the generation of loss of function (LOF) mutations in somatic cells via biallelic gene-targeting, using UAS-driven expression of the Cas9 enzyme under Gal4 control (*Port et al., 2014*). Briefly, an sgRNA (Cas9 scaffold plus guide RNA) directs Cas9 to the complementary target DNA sequence and catalyzes a double-strand break (DSB) (*Figure 1B*). Repair of this DSB occurs either by precise homology-directed repair (HDR) or more error-prone non-homologous end joining (NHEJ) (*Figure 1B*). If the targeted sequence is repaired correctly, the CRISPR machinery will target it for DSB again. If it is repaired incorrectly, this could result in small insertions or deletions (*Figure 1B*), which can cause frame-shift mutations, early stop codons, and loss of function (*Port et al., 2014*). Additionally, placing tRNA sequences between multiple sgRNAs in a single transcript allows their release by endogenous tRNA excision machinery and improves the efficiency of gene disruption (*Port and Bullock, 2016*). For example, when Port and Bullock used this strategy to express four unique sgRNAs together, ~100% of the eye area exhibited the LOF *sepia* phenotype, compared with only 11–58% from each individual sgRNA expressed alone. Thus, targeting multiple unique sgRNAs to the same gene increases the likelihood of achieving a LOF mutation in that gene (*Port and Bullock, 2016*). Finally, expressing

both the Cas9 enzyme and the sgRNA sequences from two separate UAS-transgenes reduced gene disruption in non-target tissues, likely due to the low probability of having sufficiently leaky expression of both UAS-transgenes without a Gal4 present (*Port and Bullock, 2016*).

Here, we generated UAS-transgenes expressing multiple sgRNAs that target either *timeless*, *period*, or a control gene (*acp*). We validated these constructs by showing that CRISPR-mediated gene disruption of *tim* or *per* recapitulates null mutant phenotypes when driven in all clock neurons (Tim$^+$ cells), but not in glia, and further confirmed gene disruption by qRT-PCR over the circadian cycle and brain immunostaining. We then disrupted the molecular clock in both the morning and evening oscillators (Mai179$^+$), only in the morning oscillator (Pdf$^+$), or only in the evening oscillator (Mai179$^+$Pdf$^-$). These experiments showed that, while loss of the molecular clock in both Pdf$^+$ neurons (which include the morning oscillator) and the evening oscillator neurons caused profound loss of circadian locomotor activity, loss of the molecular clock in either subset of neurons alone did not. This challenges the assumption that an internal molecular clock in the morning oscillator is required to synchronize the activity of other clock neurons and further suggests that circadian neurons act in a distributed network that can compensate for loss of the molecular clock in specific subsets.

## Results

### *UAS-sgRNA* constructs target circadian gene expression in a tissue-specific manner

We generated CRISPR tools for cell-specific gene disruption of *period* (*per*) and *timeless (tim)* (*Figure 1A*), based on previous work (*Port et al., 2014*; *Port and Bullock, 2016*). UAS-driven constructs with multiple scaffold-guide RNAs (sgRNAs) were paired with a *Gal4* expression driver and a *UAS-Cas9* construct to induce cell-specific LOF mutations (*Figure 1B*). We refer to this combination of *Gal4*-driven *UAS-sgRNA* and *UAS-Cas9* expression as '(*target gene*)$^{CRISPR}$'. In addition to *tim* and *per*, we also targeted the control gene *acp98AB* (*acp*). Because *acp* is expressed exclusively in male accessory gland cells and the testes (*Wolfner et al., 1997*; *Gelbart and Emmert, 2013*), CRISPR-mediated mutation of this gene in neurons serves as a control for any nonspecific effects due to double-strand DNA break events, such as cell death. To clone the *UAS-sgRNA* lines, we used the *pCFD6* plasmid designed by Port and Bullock (*Figure 1C*) (*Port and Bullock, 2016*). The cassette contains four unique gRNA sequences (see Materials and methods) that target the first four exons of the gene of interest to ensure efficient and specific gene disruption (*Figure 1D*).

To determine which circadian neurons require *per* and *tim* expression to influence behavioral rhythmicity, we used three previously characterized *Gal4* drivers that express in clock neurons. *Tim-Gal4* drives expression in all clock gene-expressing cells in the body, including all ~150 clock neurons (*Kaneko et al., 2000*) (*Figure 1E*). *Mai179-Gal4* drives expression in a distinct subset of clock neurons that include both morning and evening oscillator neurons: s-LNvs, 5$^{th}$ s-LNv, and 3 CRY$^+$ LNds, with weak and variable expression in DN1s and l-LNvs (*Siegmund and Korge, 2001*). *Pdf-Gal4* drives expression in the s- and l-LNvs, which express the circadian neurotransmitter *Pdf* (*Park et al., 2000*) and include the morning oscillator (*Stoleru et al., 2004*; *Grima et al., 2004*; *Guo et al., 2014*).

### CRISPR-mediated disruption of *per* or *tim* in all *tim*-expressing cells causes complete loss of behavioral and molecular rhythmicity

To test our *UAS-sgRNA* constructs, we expressed each with *UAS-Cas9* in all Tim$^+$ cells using the *tim-Gal4* driver and measured circadian locomotor activity. Flies were entrained in light/dark (LD) conditions and then shifted to constant darkness (DD) to monitor endogenous circadian locomotor activity. We found that CRISPR-targeting *per* or *tim* in Tim$^+$ cells (*tim-Gal4>per$^{CRISPR}$* or *tim-Gal4>tim$^{CRISPR}$*) led to complete loss of rhythmic behavior in DD (0% rhythmicity, *Figure 2A–D*; loss of rhythm power, *Figure 2—figure supplement 1A*), though the flies still display rhythmic behavior in LD (*Figure 2—figure supplement 2*). Thus, CRISPR-mediated disruption of *tim* or *per* in Tim$^+$ cells, which includes all clock neurons, faithfully recapitulated *tim* and *per* null mutant phenotypes (*Sehgal et al., 1994*; *Konopka and Benzer, 1971*). Control flies (*tim-Gal4>acp$^{CRISPR}$*) maintained circadian locomotor activity (94% rhythmic, 24.48 hr period; *Figure 2B,C*), indicating that nonspecific effects from UAS-Cas9 expression or CRISPR-induced double stranded breaks in Tim$^+$ cells did not

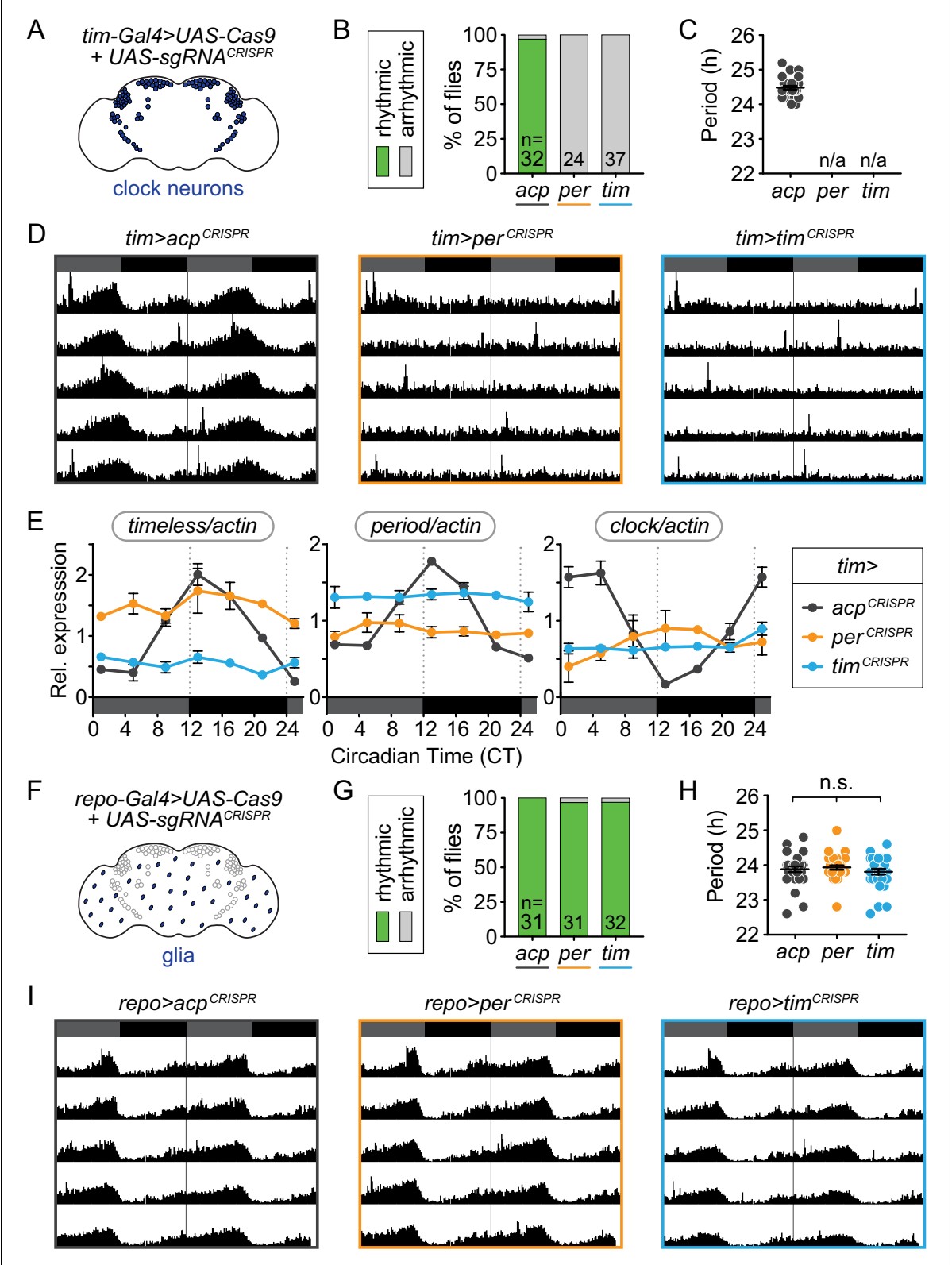

**Figure 2.** Cell-specific disruption of *per* or *tim* in circadian (Tim[+]) cells but not glial (Repo[+]) cells causes loss of behavioral rhythmicity. (A) Diagram of clock neurons targeted for CRISPR-mediated gene disruption using *tim-Gal4*. (B) Disruption of *per* or *tim* in all clock neurons caused complete loss of behavioral rhythmicity. (C) Scatter plot showing the period of rhythmic flies with *tim-Gal4*-driven disruption of *acp*, *per*, or *tim*. (D) Actograms showing average activity in constant darkness of flies with *tim-Gal4*-driven disruption of *acp*, *per*, or *tim*. Activity data is double-plotted, with six days of activity

*Figure 2 continued on next page*

Figure 2 continued

displayed. Dark gray rectangles = subjective day, black rectangles = subjective night. (E) Relative mRNA levels, measured by qRT-PCR over a 24-hour period, of the core circadian genes *timeless* (left), *period* (middle), and *clock* (right) in heads of *tim-Gal4* CRISPR flies. JTK analysis revealed that only *acp*-targeted flies display statistically significant rhythmic cycling indicative of circadian oscillation of all three genes. (F) *repo-Gal4* targets most glia for CRISPR-mediated gene disruption. (G) *repo-Gal4*-driven, CRISPR-mediated gene disruption in glia had no effect on behavioral rhythmicity. (H) Scatter plot showing the period of rhythmic flies with *repo-Gal4*-driven disruption of *acp*, *per*, or *tim*. (I) Actograms show average activity of flies in constant darkness with *repo-Gal4*-driven disruption of *acp*, *per*, or *tim* in glia.

DOI: https://doi.org/10.7554/eLife.48308.003

The following figure supplements are available for figure 2:

**Figure supplement 1.** Cell-specific disruption of *per* or *tim* in circadian (Tim[+]) cells but not glial (Repo[+]) cells causes loss of rhythm strength.
DOI: https://doi.org/10.7554/eLife.48308.004
**Figure supplement 2.** Average actograms for *tim-Gal4* and *repo-Gal4* targeted flies under 12 hr:12 hr light:dark conditions.
DOI: https://doi.org/10.7554/eLife.48308.005
**Figure supplement 3.** Rhythmicity analysis of control flies.
DOI: https://doi.org/10.7554/eLife.48308.006
**Figure supplement 4.** CRISPR-targeting of *per* or *tim* in Tim[+] neurons leads to overall loss of Per and Tim protein.
DOI: https://doi.org/10.7554/eLife.48308.007
**Figure supplement 5.** CRISPR-targeting of *per* or *tim* in Repo[+] glia leads to overall loss of Per protein in glia but not neurons.
DOI: https://doi.org/10.7554/eLife.48308.008

cause loss of rhythms. Control flies carrying individual components of the CRISPR-mediated deletion (*tim-Gal4* driver, *UAS-gRNA*, and *UAS-Cas9* lines) were also highly rhythmic (*Figure 2—figure supplement 3*). Together, our results suggest that CRISPR-targeting of *tim* and *per* results in complete functional ablation of the molecular clock, in contrast to the lengthened rhythms sometimes observed with RNAi, which are thought to reflect incomplete reduction of gene expression (*Martinek and Young, 2000*; *Young, 1998*).

To confirm that rhythmic transcription of circadian clock genes is disrupted by CRISPR-targeting *tim* or *per* in Tim[+] cells, we analyzed mRNA from fly heads collected over the circadian cycle. In wild-type fly heads, clock gene mRNA levels oscillate with approximately 24-hour periodicity in constant darkness (*Sehgal et al., 1994*; *Sehgal et al., 1995*; *Hardin et al., 1990*). We found that control flies (*tim-Gal4>acp*[CRISPR]) also displayed robust and statistically significant oscillation of *timeless*, *period*, and *clock* transcripts (*Figure 2E*, gray). In contrast, CRISPR-targeting of *tim* or *per* in Tim[+] cells resulted in arrhythmic transcription of all three molecular clock genes (*Figure 2E*, orange and blue), consistent with the behavioral arrhythmicity caused by these manipulations (*Figure 2B*). Furthermore, *tim* transcript levels in *tim-Gal4>tim*[CRISPR] flies and *per* transcript levels in *tim-Gal4>per*[CRISPR] flies were reduced to levels similar to the lowest baseline levels for these transcripts in control flies. We note that *tim* transcripts, though arrhythmic, were elevated after disruption of *per* (*tim-Gal4 >per*[CRISPR] flies) and vice versa for *per* transcripts after disruption of *tim*. These results are consistent with earlier findings indicating that loss of either Per or Tim, inhibitors of Clock/Cycle, causes constitutive activity of the Clock/Cycle transcription complex and elevated levels of *per* or *tim* transcripts (*Allada et al., 1998*; *Darlington et al., 1998*).

To further confirm the efficiency of our gene disruption, we performed immunofluorescence analysis on the brains of CRISPR-targeted flies (*tim-Gal4>gene*[CRISPR]) for Per and Tim at ZT0, along with *per*[01] null mutants (*Figure 2—figure supplement 4A,B*). At ZT0, Per and Tim proteins are highly expressed and localized to the nucleus in wild-type flies (*Vosshall et al., 1994*; *Darlington et al., 1998*; *Price et al., 1995*). Consistent with this, control flies (*tim-Gal4>acp*[CRISPR]) exhibited high levels of Per and Tim protein expression, colocalized in the nucleus. In contrast, in flies CRISPR-targeted for *per* or *tim* in Tim[+] cells (*tim-Gal4>per*[CRISPR] and *tim-Gal4>tim*[CRISPR]), we observed loss of nuclear Per or Tim staining, similar to *per*[01] null mutants (*Figure 2—figure supplement 4A,B*). CRISPR-targeting of *per* led to loss of Per signal and only cytoplasmic Tim signal; CRISPR-targeting of *tim* led to loss of both Tim and Per signal, presumably because Per is unstable without Tim (*Figure 2—figure supplement 4C*) (*Price et al., 1995*; *Price et al., 1998*). Taken together, these results show that CRISPR-mediated, *Gal4*-driven disruption of *per* and *tim* in Tim[+] cells is highly efficient on both the mRNA and protein levels and is sufficient to block locomotor activity rhythms.

## CRISPR-mediated disruption of *per* or *tim* in glia (Repo⁺ cells) does not disrupt behavioral rhythmicity

As a second control for the effect of CRISPR-induced DNA damage and to confirm that this CRISPR gene targeting is Gal4-specific, we CRISPR-targeted *tim* and *per* in Repo⁺ glia, using the glial driver *repo-Gal4* (*Halter et al., 1995*; *Xiong et al., 1994*). Glia are not predicted to control circadian locomotor activity via circadian clock gene expression (*Ng et al., 2011*). We found that nearly all flies, whether CRISPR-targeted for *tim*, *per*, or *acp*, were highly rhythmic (100% of *repo-Gal4>acp^CRISPR* and 97% of *repo-Gal4>per^CRISPR* and *tim^CRISPR*) (*Figure 2F–I*). CRISPR-targeting *per* or *tim* in Repo⁺ cells did not reduce rhythm strength or affect rhythmic behavior in LD relative to the *acp*-targeted control (*Figure 2—figure supplements 1B* and *2*). We also confirmed through immunofluorescence analysis that the CRISPR-mediated deletion was both efficient in (*Figure 2—figure supplement 5A*) and specific to (*Figure 2—figure supplement 5B*) glial cells. These results demonstrate that there is no leaky or non-Gal4-mediated expression of both the *UAS-Cas9* and *UAS-sgRNA* that affects rhythmicity. These results further confirm previously published results that, while circadian locomotor activity requires intact glial cells, it does not require glial expression of clock genes (*Ng et al., 2011*).

## Disruption of *per* or *tim* in both morning and evening oscillators (Mai179⁺ neurons) causes complete loss of circadian locomotor activity

To test the effect of disrupting *per* or *tim* in both the morning and evening oscillators, we expressed our *UAS-sgRNA* constructs in the s-LNvs (including the 5ᵗʰ s-LNv) and 3 CRY⁺ LNds, with weak or variable expression in l-LNvs, DN1s, and non-clock neurons, using the *Mai179-Gal4* driver (*Figure 3A*) (*Grima et al., 2004*; *Yao et al., 2016*; *Siegmund and Korge, 2001*; *Rieger et al., 2009*). We found that 100% of flies CRISPR-targeted for *per* and *tim* in Mai179⁺ cells (*Mai179-Gal4> per^CRISPR* and *Mai179-Gal4> tim^CRISPR*) were arrhythmic, while 91% of control flies (*Mai179-Gal4> acp^CRISPR*) remained rhythmic (*Figure 3A–D*; *Figure 3—figure supplement 1*). Thus, the molecular clock is required in Mai179⁺ neurons for circadian locomotor activity.

To confirm the loss of protein after gene disruption, we measured Per and Tim protein levels in Mai179⁺ neurons. We co-immunostained brains for Per and Tim and quantified nuclear fluorescence intensity at ZT0. We found that control flies showed robust nuclear staining of both Per and Tim at ZT0 (*Figure 3E–H* in gray; *Figure 3—figure supplement 2*), whereas *per* disruption in Mai179⁺ neurons caused near-complete loss of Per protein, as measured by the number of Per-positive nuclei per brain and average Per fluorescence intensity per brain (*Figure 3E and G*, orange dots). In *per*-targeted flies, Tim protein remained mostly cytoplasmic (*Figure 3—figure supplement 2*) (*Vosshall et al., 1994*; *Price et al., 1995*). These results suggest near-complete CRISPR-mediated *per* gene disruption in *Mai179-Gal4> per^CRISPR* flies, consistent with the observed complete loss of behavioral rhythmicity (*Figure 3A–D*). For *Mai179*-specific *tim*-targeted flies (*Mai179-Gal4> tim^CRISPR*), only a small number of nuclei displayed Tim intensity levels close to the levels observed in control nuclei (*Figure 3F*, compare blue to gray), while the average nuclear fluorescence of Tim is near zero (*Figure 3H*). Further analysis of these brains revealed that most of the minority of neurons with incomplete loss of Tim/Per protein were large ventral lateral neurons (l-LNvs); deletion was highly efficient in s-LNvs and LNds (*Figure 3—figure supplement 3*). Additionally, Per nuclear staining was nearly eliminated in *tim*-targeted flies (*Figure 3E,G*). Thus, our results support robust disruption of molecular clock function in these neurons, consistent with the observed complete loss of behavioral rhythmicity.

## Disruption of *per* or *tim* in the morning oscillator (Pdf⁺ s-LNv neurons) does not cause loss of circadian locomotor activity

To investigate the role of the circadian clock in morning oscillator neurons alone, we next induced CRISPR-mediated gene disruption of *tim* and *per* in *Pdf*-expressing cells using *Pdf-Gal4*. Pigment-dispersing factor (Pdf) is a neuropeptide expressed and secreted by the l-LNv and s-LNv neurons, which form the morning oscillator (*Stoleru et al., 2004*; *Renn et al., 1999*; *Helfrich-Förster, 1995*; *Rieger et al., 2009*) (*Figure 4A*). While Pdf⁺ neurons are thought to be essential for circadian locomotor activity, we found that CRISPR-targeting of *per* and *tim* in Pdf⁺ neurons did not cause complete loss of rhythmicity. 74% of *Pdf*-specific, *per*-targeted flies (*Pdf-Gal4>per^CRISPR*) and 83% of

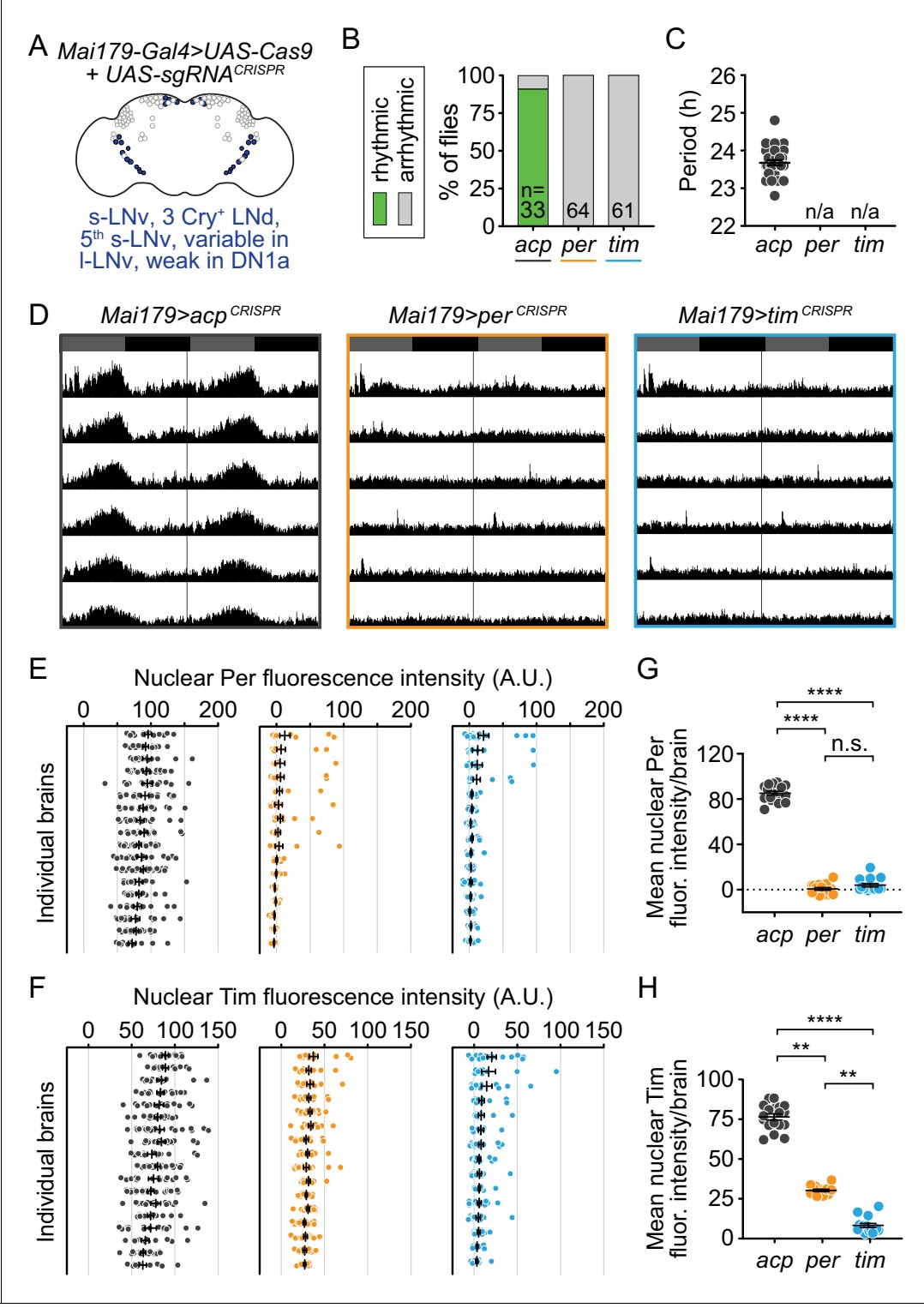

**Figure 3.** Cell-specific disruption of *per* or *tim* in Mai179+ neurons causes complete loss of behavioral rhythmicity and efficient loss of the targeted protein. (A) Diagram of the circadian neuron subset marked by *Mai179-Gal4*. (B) Disruption of *per* or *tim* in Mai179+ neurons caused complete loss of behavioral rhythmicity. (C) Scatter plot showing the period of rhythmic flies with *Mai179-Gal4*-driven disruption of *acp*, *per*, or *tim*. (D) Average actograms showing the activity of flies in constant darkness with *Mai179-Gal4*-driven disruption of *acp*, *per*, or *tim*. Six days of activity are displayed, double-plotted. Dark gray rectangles = subjective day, black rectangles = subjective night. (E and F) Background-subtracted nuclear fluorescence intensity of Per (E) or Tim (F) at ZT0 in GFP+ neurons, grouped by brain with mean ± SEM shown. Individual brains are shown in the same order in both E and F. (G and H) Mean nuclear fluorescence intensity of Per (G) or Tim (H) at ZT0 in GFP+ neurons, averaged per brain (*acp* n = 18; *per* n = 16; *tim*

*Figure 3 continued on next page*

Figure 3 continued

n = 15). **: p<0.01; ****: p<0.0001; n.s.: not significant, p>0.05. Significance determined by Kruskal-Wallis nonparametric ANOVA (to account for non-normality of samples) followed by Dunn's multiple comparisons test; reported p-values are multiplicity adjusted to account for multiple comparisons.
DOI: https://doi.org/10.7554/eLife.48308.009

The following figure supplements are available for figure 3:

**Figure supplement 1.** Cell-specific disruption of *per* or *tim* in Mai179+ cells causes loss of rhythm strength.
DOI: https://doi.org/10.7554/eLife.48308.010

**Figure supplement 2.** Period and Timeless immunohistochemistry in *Mai179-Gal4*-driven CRISPR flies.
DOI: https://doi.org/10.7554/eLife.48308.012

**Figure supplement 3.** Cell-specific disruption of *per* or *tim* in Mai179+ neurons is most efficient in dorsal lateral (LNd) and small ventral lateral (s-LNv) neurons.
DOI: https://doi.org/10.7554/eLife.48308.013

**Figure supplement 4.** Average actograms for *Mai179-Gal4* targeted flies under 12 hr:12 hr light:dark conditions.
DOI: https://doi.org/10.7554/eLife.48308.011

*Pdf*-specific *tim*-targeted flies (*Pdf-Gal4>tim^CRISPR*) were rhythmic, as compared to 100% of controls (*Pdf-Gal4>acp^CRISPR*) (**Figure 4B**).

Qualitatively, activity rhythms of individual flies often appeared more ambiguous, and therefore some were scored as 'weakly rhythmic' (**Figure 4B**; **Figure 4—figure supplement 1**); indeed the rhythm strength is significantly reduced, relative to *acp*-targeted controls, though not completely abolished as seen in *tim-Gal4*- and *Mai179-Gal4*-driven CRISPR targeting (**Figure 4—figure supplement 2**). We confirmed the results for overall rhythmicity by an automated scoring method, Lomb-Scargle periodogram analysis (see Materials and methods). Again, *Pdf*-specific targeting of *per* or *tim* resulted in mostly rhythmic flies (87% and 92%, respectively), similar to controls (95% rhythmic) (**Supplementary file 1**). All flies were tracked for 9–10 days after shifting to constant darkness, because *Pdf* mutants and *Pdf receptor* (*Pdfr*) mutants lose their rhythms only after 1–3 days in constant darkness (**Renn et al., 1999**). We classified flies as rhythmic if they maintained activity rhythms for the entire 9–10 days. Finally, though the morning oscillator is thought to delay the evening peak of activity and thus control period length (**Yao et al., 2016**), the average period length of *Pdf*-specific *per* or *tim*-targeted flies (23.87 and 23.42 hr, respectively) was similar to controls (23.88 hr) (**Figure 4C**). These results suggest that the molecular clock is not required in the morning oscillator (Pdf+ s-LNv neurons) for overall circadian locomotor activity nor to control the pacing of the evening oscillator.

Pdf+ s-LNv neurons also regulate 'morning anticipation,' or increased activity just before the transition to lights-on (**Stoleru et al., 2004**; **Grima et al., 2004**). The evening oscillator neurons regulate 'evening anticipation,' or increased activity just before the transition to lights off. To determine whether loss of the molecular clock in Pdf+ neurons specifically affects morning anticipation, we analyzed both types of anticipation in *Pdf*-specific *per* and *tim*-targeted flies relative to controls. While evening anticipation was intact after CRISPR-targeting of *tim* or *per* in Pdf+ neurons, morning anticipation was absent or diminished (**Figure 4E**). To quantify this, we calculated morning and evening anticipation indices (MAIs and EAIs) for each genotype (see Materials and methods). The MAI in DD day two was significantly reduced after targeting of *per* or *tim* in Pdf+ neurons (*Pdf-Gal4>per^CRISPR* and *Pdf-Gal4>tim^CRISPR*) relative to controls (*Pdf-Gal4>acp^CRISPR*) (**Figure 4F**), whereas EAI was not reduced (**Figure 4G**). In LD, while the overall activity pattern appears normal (**Figure 4—figure supplement 3**), the MAI was reduced in *Pdf-Gal4>per^CRISPR* flies, though not in *Pdf-Gal4>tim^CRISPR* flies, and the EAI was unaffected (**Figure 4—figure supplement 4A–C**). These phenotypes were similar on days 3–9 in DD (**Figure 4—figure supplement 4D–F**). These results suggest that, while the molecular clock in Pdf+ neurons is not required for locomotor rhythmicity, it is required for morning anticipatory behavior.

## CRISPR-mediated disruption of *per* or *tim* in Pdf+ neurons causes near-complete loss of Per and Tim protein

If the *Pdf-Gal4* driver is not strong enough to fully disrupt *per* or *tim* in Pdf+ neurons, this could result in an incomplete behavioral phenotype. To confirm that *per* and *tim* disruption in Pdf+ cells is as efficient as observed with *tim-Gal4* and *Mai179-Gal4*, which caused arrhythmicity, we performed

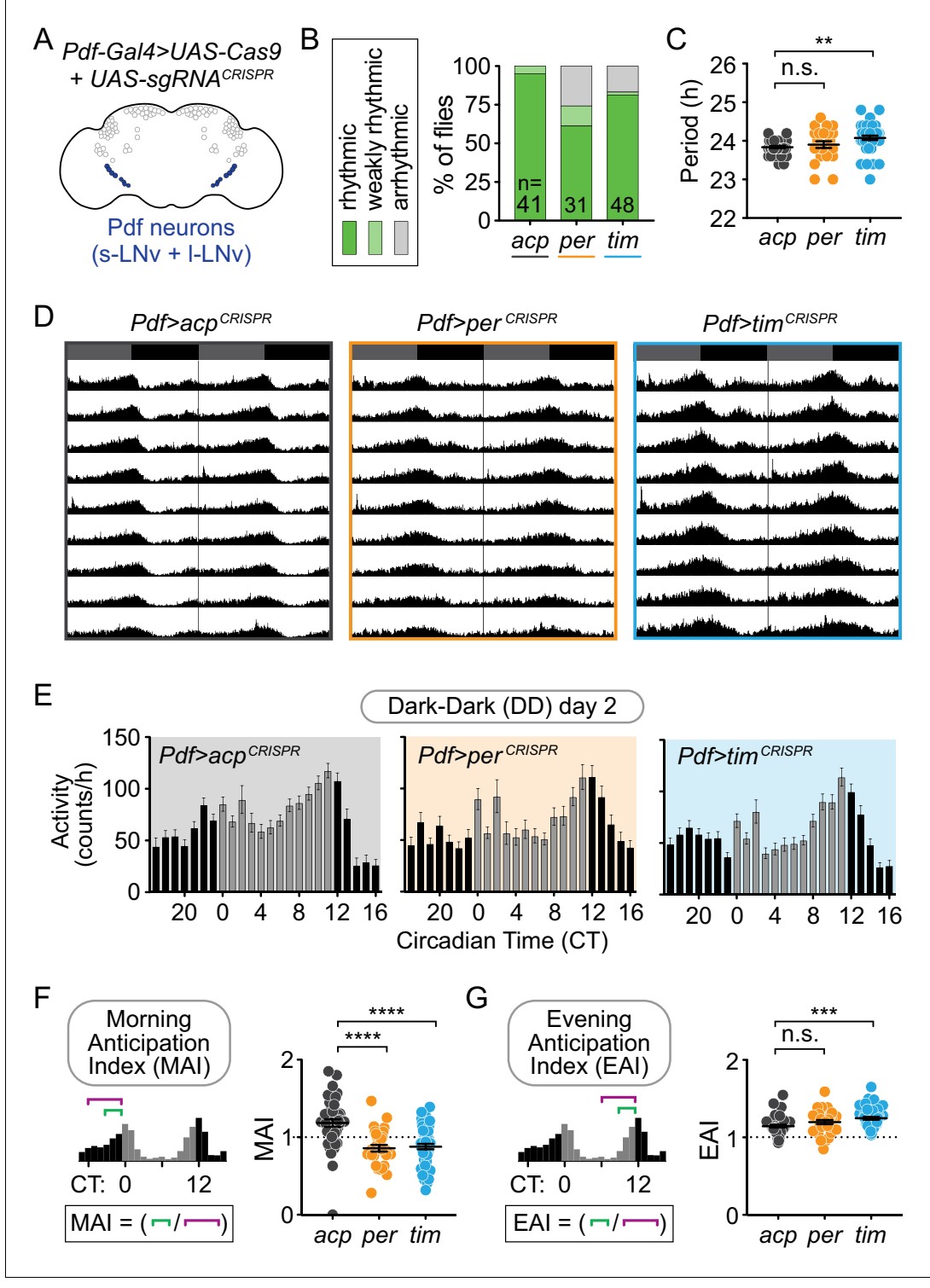

**Figure 4.** Cell-specific disruption of *per* or *tim* in Pdf[+] neurons causes incomplete loss of behavioral rhythmicity and loss of morning anticipation in constant darkness. (**A**) Diagram showing Pdf[+] circadian neurons. (**B**) CRISPR-mediated disruption of *per* or *tim* in Pdf[+] neurons using the *Pdf-Gal4* driver caused incomplete loss of behavioral rhythmicity. (**C**) Scatter plot showing the period of rhythmic flies with *Pdf-Gal4*-driven disruption of *acp*, *per*, or *tim*. (**D**) Actograms showing average activity of flies in constant darkness with *Pdf-Gal4*-driven disruption of *acp*, *per*, or *tim*. Nine days of activity are displayed, double-plotted. Dark gray rectangles = subjective day, black rectangles = subjective night. (**E**) Average hourly activity counts during the second day of complete darkness (DD Day 2; gray bars = CT 0–11, black bars = CT 12–23). Mean number of beam breaks per hour is shown ± SEM. (**F**)

*Figure 4 continued on next page*

*Figure 4 continued*

Morning Anticipation Index (MAI) was calculated by dividing the average hourly activity for CT 21–23 by the average hourly activity for CT 18–23. (G) Evening Anticipation Index (EAI) was calculated by dividing the average hourly activity for CT 9–11 by the average hourly activity for CT 6–11. For scatter plots, each point represents an individual fly and mean ± SEM is shown; ***: p<0.001; ****: p<0.0001; n.s.: not significant, p>0.05. Significance determined by Kruskal-Wallis nonparametric ANOVA (to account for non-normality of samples) followed by Dunn's multiple comparisons test; reported p-values are multiplicity adjusted to account for multiple comparisons.

DOI: https://doi.org/10.7554/eLife.48308.014

The following figure supplements are available for figure 4:

**Figure supplement 1.** Representative actograms for each phenotypic class.

DOI: https://doi.org/10.7554/eLife.48308.017

**Figure supplement 2.** Cell-specific disruption of *per* or *tim* in Pdf[+] cells causes reduction in rhythm strength.

DOI: https://doi.org/10.7554/eLife.48308.015

**Figure supplement 3.** Average actograms for *Pdf-Gal4* targeted flies under 12 hr:12 hr light:dark conditions.

DOI: https://doi.org/10.7554/eLife.48308.016

**Figure supplement 4.** Cell-specific disruption of *per* or *tim* in Pdf[+] neurons causes loss of the morning anticipatory peak under constant conditions.

DOI: https://doi.org/10.7554/eLife.48308.018

quantitative immunofluorescence analysis of Per and Tim protein levels (*Figure 5*). Control flies (*Pdf-Gal4>acp[CRISPR]*) displayed the expected robust nuclear staining of Per and Tim in both the s-LNvs and l-LNvs (*Figure 5A,D–G*, in gray). In contrast, *Pdf*-specific *per*-targeted flies (*Pdf-Gal4>per[CRISPR]*) exhibited a near-complete absence of nuclear Per immunofluorescence signal in both s-LNvs and l-LNvs (*Figure 5A,D,F*; *Figure 5—figure supplement 1*). Similar to what we observed in the *Mai179*-specific *per* disruption, any remaining Tim signal was localized to the cytoplasm (*Figure 5A*). In *Pdf*-specific *tim*-targeted flies (*Pdf-Gal4>tim[CRISPR]*), we observed a similar near-complete reduction in Tim protein levels in LNvs, with relatively few Tim[+] nuclei remaining and an average fluorescence intensity per brain near zero (*Figure 5A,E,G*). CRISPR-targeting of *tim* also resulted in near-complete loss of Per protein, indistinguishable from loss of Per in *per*-targeted flies (*Figure 5F*) (*Price et al., 1995*; *Price et al., 1998*). These results suggest that the persistence of circadian locomotor activity seen in *Pdf*-specific *per* and *tim*-targeted flies is not due to incomplete disruption of the targeted gene. Taken together, our behavioral and quantitative immunofluorescence analysis suggest that the molecular clock in Pdf[+] clock neurons is not required for circadian locomotor activity.

We also used a *UAS-myr-GFP* to label the membranes of Pdf[+] neurons and counted the number of GFP[+] cells in each brain to confirm that the CRISPR-induced DNA damage in our system does not cause cell death. There are 8 Pdf[+] LNvs in each hemisphere, totaling 16 neurons in each fly brain (*Lin et al., 2004*). We found no significant difference in the number of GFP[+] LNvs in each brain between experimental flies and controls (*Figure 5C*). This result indicates that CRISPR-mediated gene disruption in Pdf[+] neurons does not cause significant cell death.

## Restriction of CRISPR-mediated disruption of *per* or *tim* to evening oscillator (Mai179[+]Pdf[-]) neurons by blocking disruption in Pdf[+] neurons restores behavioral rhythmicity

To determine whether the molecular clock is required in evening oscillator neurons, we paired the *Mai179-Gal4* driver with *Pdf-Gal80*, blocking CRISPR-targeting of *tim* or *per* in Pdf[+] neurons. This Gal80-mediated inhibition of Gal4 restricts CRISPR-targeting to the evening oscillator (Mai179[+] Pdf[-] neurons): the 5th sLNv, CRY[+] LNds, and a small subset of DN1s (*Figure 6A*). We found that overall locomotor rhythmicity was maintained; 82% of *per*-targeted and *tim*-targeted flies (*Mai179-Gal4/Pdf-Gal80>per[CRISPR]* and *Mai179-Gal4/Pdf-Gal80>tim[CRISPR]*) were rhythmic, similar to 89% of control flies (*Figure 6B–D*). The rhythm strength of *tim*-targeted flies was slightly reduced relative to *acp* controls, whereas rhythm strength in *per*-targeted flies was unaffected (*Figure 6—figure supplement 1*). We also verified the efficiency and specificity of the deletion in Mai179[+] Pdf[-] neurons through immunofluorescence analysis (*Figure 6E,F*; *Figure 6—figure supplement 2*). The CRISPR deletion was highly efficient in RFP[+] LNds and the 5th s-LNv, representing the minimal evening

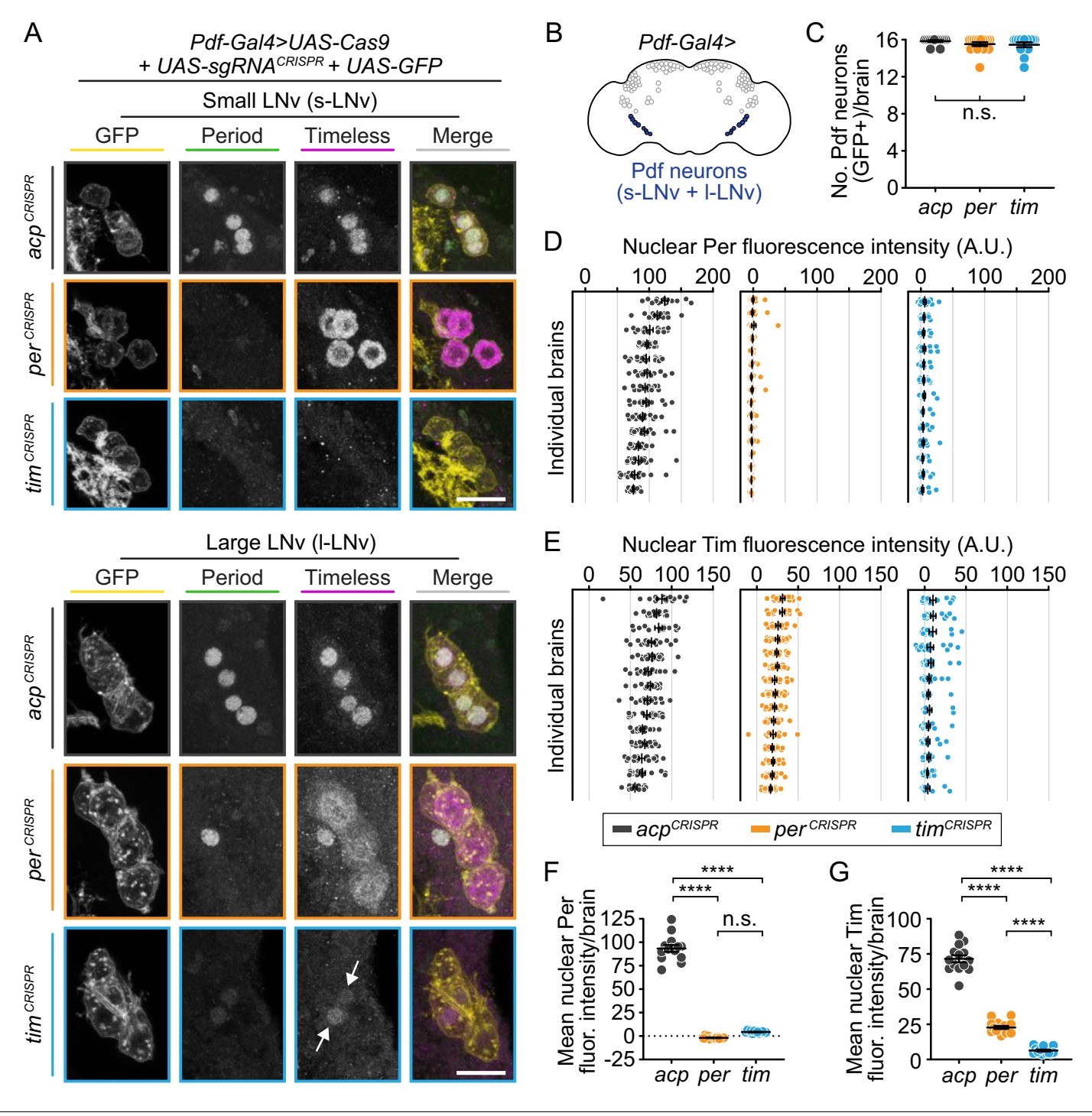

**Figure 5.** CRISPR-mediated disruption of *per* or *tim* in Pdf[+] neurons leads to efficient loss of the targeted protein. (**A**) Maximum intensity projections of *Pdf-Gal4*-driven GFP[+] small and large ventral lateral neurons (s- and l-LNv) with immunohistochemistry for GFP (yellow), Period (green) and Timeless (magenta) at ZT0. Scale bar = 10 μm; arrows indicate CRISPR-targeted nuclei with residual protein signal. (**B**) Diagram showing *Pdf-Gal4* expression in the 4 large and four small ventral lateral neurons (l- and s-LNv), bilaterally. (**C**) Quantification of the number of GFP[+] neurons per brain (*acp* n = 14; *per* n = 15; *tim* n = 13 brains). (**D and E**) Background-subtracted nuclear fluorescence intensity of Per (**D**) or Tim (**E**) at ZT0 in GFP[+] neurons, grouped by brain with mean ± SEM shown. (**F and G**) Mean nuclear fluorescence intensity of Per (**G**) or Tim (**H**) at ZT0 in GFP[+] neurons, averaged per brain (*acp* n = 14; *per* n = 16; *tim* n = 14 brains). ****: p<0.0001; n.s.: not significant, p>0.05. Significance was determined by one-way ANOVA followed by Tukey's multiple comparisons test; reported p-values are multiplicity adjusted to account for multiple comparisons.

DOI: https://doi.org/10.7554/eLife.48308.019

*Figure 5 continued*

The following figure supplement is available for figure 5:

**Figure supplement 1.** Cell-specific disruption of *per* or *tim* in Pdf+ neurons has similar efficiency in small ventral lateral (s-LNv) and large ventral lateral (l-LNv) neurons.

DOI: https://doi.org/10.7554/eLife.48308.020

oscillator (*Figure 6E*; *Figure 6—figure supplement 2*). We found that the *PdfGal80* construct largely protected Pdf+ neurons from CRISPR targeting, as evidenced by wild-type levels of Per signal in the majority of Pdf+ cells. There was a minority of LNvs that lost Per expression; this incomplete protection was slightly more prevalent in *tim*-targeted flies (*Figure 6F*; *Figure 6—figure supplement 2*), which may explain the slightly reduced rhythm strength in *tim*-targeted but not *per*-targeted flies (*Figure 6—figure supplement 1*). The result that restoring molecular clock expression in Pdf+ neurons rescued overall rhythmicity is consistent with previous studies in which rhythmicity was restored by expression of *UAS-per* in Pdf+ neurons of *per* null mutant flies (*Grima et al., 2004*). Thus, while *per* and *tim* expression are not necessary in Pdf+ neurons for rhythmicity (*Figure 4*), their expression in Pdf+ and Mai179- neurons is sufficient for circadian locomotor activity. Taken together, our results suggest that the molecular clock may be sufficient in either the morning oscillator or evening oscillator for circadian locomotor activity and that the molecular clock must be disrupted in both oscillators to disrupt circadian locomotor activity.

Because Mai179+ Pdf- neurons comprise the minimal evening oscillator, we also measured 'evening anticipation,' or increased activity just before the transition to lights off. Similar to our observation that *per* or *tim* disruption in the Pdf+ morning oscillator led to a loss of morning anticipation, *per* or *tim* disruption in the Mai179+ Pdf- evening oscillator neurons led to a loss of evening anticipation activity (*Figure 6—figure supplement 3*). The EAI was significantly reduced in evening oscillator-specific *per*-targeted flies and the EAI of *tim*-targeted flies was trending, but not significantly reduced ($p<0.10$), relative to controls. The morning anticipation indices (MAI) remained intact and were not significantly different from controls (*Figure 6—figure supplement 3*), further demonstrating that *per* or *tim* expression in Pdf+ s-LNv morning oscillator neurons is both necessary and sufficient for morning anticipatory activity.

## Discussion

Our understanding of how circadian neurons communicate with each other to control locomotor rhythmicity is still evolving. Over a decade ago, some of the first evidence was presented to support a 'dual oscillator' model in which Pdf+ s-LNvs are classified as 'morning cells' that control morning anticipation and drive the maintenance of overall rhythmicity. This model also suggests that Pdf+ s-LNvs dominate over other circadian neurons, such as 'evening cells' (*Top and Young, 2018*; *Stoleru et al., 2004*; *Grima et al., 2004*; *Guo et al., 2014*). More recent evidence has questioned this hierarchical model and instead suggests a complex network in which the control of circadian behavior is distributed among many subgroups of neurons (*Yao et al., 2016*; *Yao and Shafer, 2014*; *Stoleru et al., 2005*; *Zhang et al., 2010*; *Bulthuis et al., 2019*). Most of these recent studies utilized proteins known to alter period length when expressed ubiquitously (mutant kinases, mutated kinase targets, or dominant negative constructs). When these proteins were overexpressed in specific clock neurons such as Pdf+ neurons or evening oscillators, they exerted varying levels of control over the molecular clocks in other neurons and overall circadian locomotor activity. While these studies were elegantly done using available tools, overexpression studies carry the potential problem of gain of function. Moreover, proteins that regulate the core molecular clock have significantly different roles in different clock neurons (*Top et al., 2016*; *Top et al., 2018*). The best genetic tools are those that cause loss of function. Here we developed and validated tools for CRISPR-mediated disruption of the molecular clock in targeted subsets of circadian neurons.

Our results demonstrate the efficacy and utility of genetic constructs that mediate tissue-specific CRISPR-targeting of two key circadian clock genes: *timeless* and *period*. We showed that these constructs recapitulate known mutant phenotypes, such as complete loss of locomotor activity rhythms when driven in all *tim*-expressing cells (*Figure 2*). We validated the extent of gene disruption at both the mRNA and protein levels (*Figures 2*, *3*, *5* and *6*) and showed that gene targeting effects are

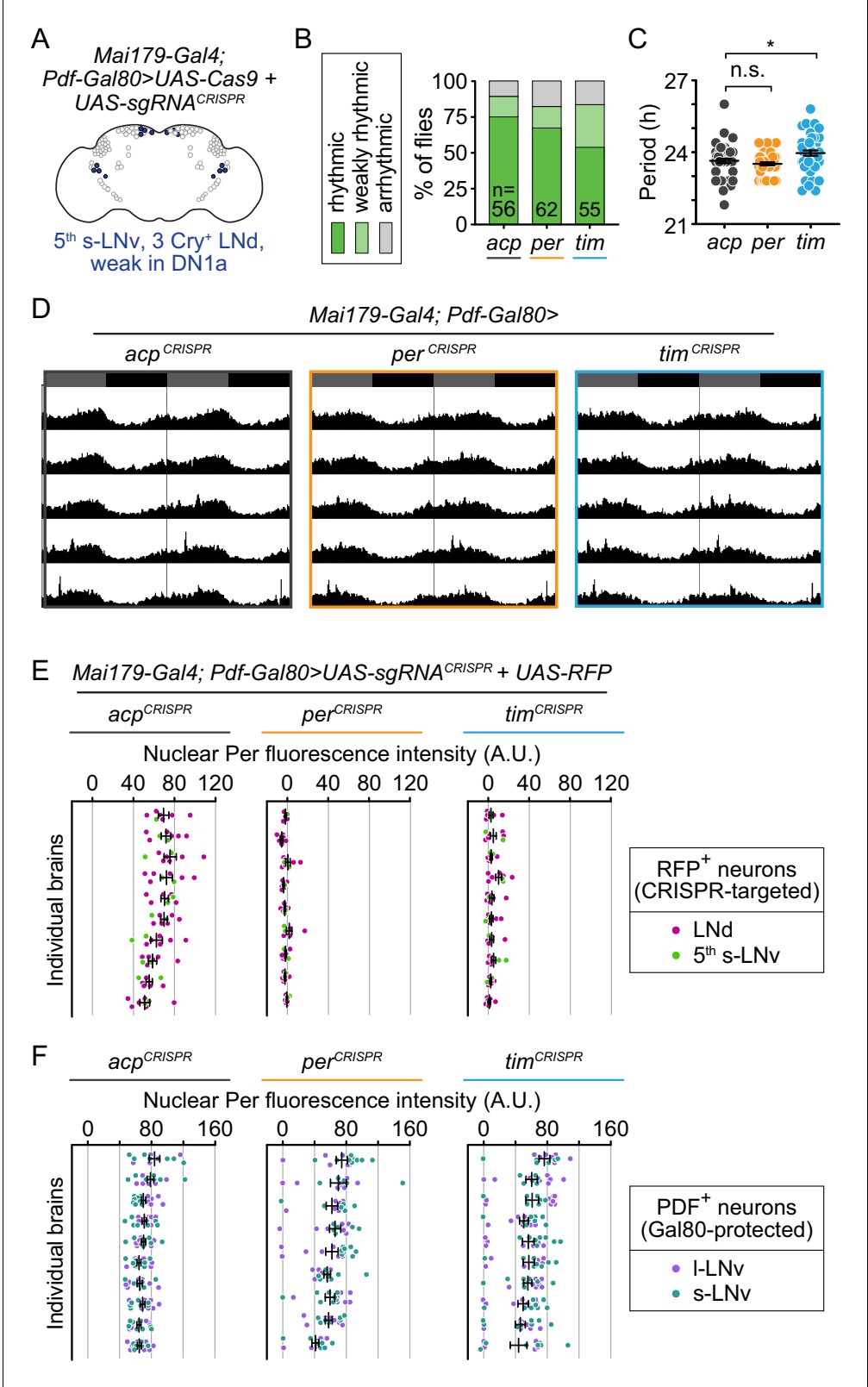

**Figure 6.** Cell-specific disruption of *per* or *tim* in Mai179+Pdf − neurons does not affect behavioral rhythmicity. (**A**) Diagram showing circadian neurons targeted using *Mai179-Gal4; Pdf-Gal80*. (**B**) CRISPR-mediated disruption of *per* or *tim* in Mai179+Pdf− neurons did not affect overall rhythmicity. (**C**) Scatter plot showing the period of rhythmic flies with *Mai179-Gal4; Pdf-Gal80*-driven disruption of *acp*, *per*, or *tim*. (**D**) Average actograms showing

*Figure 6 continued on next page*

*Figure 6 continued*

the activity of flies in constant darkness with *Mai179-Gal4; Pdf-Gal80*-driven disruption of *acp*, *per*, or *tim*. Six days of activity are displayed, double-plotted. Dark gray rectangles = subjective day, black rectangles = subjective night. (**E and F**) Background-subtracted nuclear fluorescence intensity of Per at ZT0 in (**E**) RFP$^+$ LNds (magenta) and the 5$^{th}$ s-LNv (light green) and (**F**) PDF$^+$ s- (purple) and l-LNv neurons (dark green), grouped by brain with mean ± SEM shown (*acp* n = 10; *per* n = 9; *tim* n = 10). *: p<0.05; n.s.: not significant, p>0.05. Significance was determined by one-way ANOVA followed by Tukey's multiple comparisons test; reported p-values are multiplicity adjusted to account for multiple comparisons.

DOI: https://doi.org/10.7554/eLife.48308.021

The following figure supplements are available for figure 6:

**Figure supplement 1.** Cell-specific disruption of *tim,* but not *per*, in Mai179$^+$Pdf$^-$ cells causes a slight reduction in rhythm strength.

DOI: https://doi.org/10.7554/eLife.48308.022

**Figure supplement 2.** *Mai179-Gal4; Pdf-Gal80*-driven CRISPR-targeting protects PDF$^+$ cells from gene disruption.

DOI: https://doi.org/10.7554/eLife.48308.024

**Figure supplement 3.** Cell-specific disruption of *per* in Mai179$^+$Pdf$^-$ neurons causes loss of the evening anticipatory peak under constant conditions.

DOI: https://doi.org/10.7554/eLife.48308.025

**Figure supplement 4.** Average actograms for *Mai179-Gal4; Pdf-Gal80* targeted flies under 12 hr:12 hr light:dark conditions.

DOI: https://doi.org/10.7554/eLife.48308.023

---

Gal4-dependent, as there is no effect on locomotor rhythmicity when *tim* or *per* are disrupted in glia, nor any effect of the CRISPR tools in the absence of a Gal4 driver. We then used these lines to dissect the molecular clock requirements in different subsets of circadian regulatory neurons. In summary, loss of *per* or *tim* expression in both the morning and the evening oscillators (Mai179$^+$) causes arrhythmicity. In contrast, loss of *per* and *tim* expression in only Pdf$^+$ cells (which include the morning oscillator) or the evening oscillator neurons (Mai179$^+$ Pdf$^-$) does not cause arrhythmicity. Thus, in contradiction of the previous paradigm, *per* and *tim* expression in Pdf$^+$ neurons is not necessary for circadian locomotor activity. It should be noted that while the molecular clock in Pdf$^+$ neurons is sufficient for rhythmicity in the context of *Mai179-Gal4/Pdf-Gal80* CRISPR-targeting, Mai179$^-$ clock neurons other than Pdf$^+$ neurons still express *per* and *tim* and may also be required for rhythmicity. Moreover, while *Pdf-Gal80* efficiently protected against CRISPR disruption driven by *Mai179-Gal4*, future experiments utilizing Gal80 protection should take into account the relative developmental timing of the Gal4 and Gal80 constructs because the CRISPR disruption event is irreversible.

While this manuscript was in preparation, we became aware of a similar study examining the requirement of the molecular clock in different subsets of clock neurons. Schlichting et al. also used a tissue-specific CRISPR-mediated mutagenesis strategy to target *period* with three gRNAs and obtained similar results. Consistent with our results, they found that disruption of *per* expression in Pdf$^+$ cells did not cause loss of circadian locomotor activity. Moreover, loss of Clock protein cycling in Pdf$^+$ neurons due to Pdf-specific neuronal silencing did not cause loss of circadian locomotor activity. Taken together, our results and those from the Rosbash lab demonstrate that the molecular clock is not required in Pdf$^+$ neurons for circadian locomotor activity and suggests that rhythmicity is a network property.

Our evidence supports a model of independent morning and evening oscillators that control their respective anticipatory behaviors and can compensate for each other to maintain overall locomotor rhythmicity. CRISPR targeting *per* or *tim* in Pdf$^+$ neurons, which contain the morning oscillator, led to a loss of morning anticipatory behavior (*Figure 4*). This is consistent with previous reports demonstrating that ablation of *Pdf*-expressing cells or loss of function of *Pdf* itself or its receptor *Pdfr* caused loss of morning anticipation (*Renn et al., 1999*; *Mertens et al., 2005*). This suggests that this specific aspect of circadian behavior, morning anticipatory activity, requires the molecular clock in Pdf$^+$ neurons. However, while *Pdf* mutants and Pdf$^+$ cell ablation led to a loss of overall rhythmicity (*Renn et al., 1999*; *Fernández et al., 2008*; *Shafer and Taghert, 2009*; *Park et al., 2000*), disruption of the molecular clock (*tim* or *per*) in Pdf$^+$ neurons did not. These results suggest that locomotor rhythmicity, while dependent on *Pdf* expression and Pdf$^+$ neurons, is not dependent on

the function of the molecular clock within Pdf⁺ neurons. Similarly, disruption of *per* or *tim* in just the evening oscillator neurons (*Mai179-Gal4/Pdf-Gal80*), led to a loss of evening anticipatory behavior, but not locomotor rhythmicity (*Figure 6*). This also demonstrated that while an intact molecular clock in morning oscillator neurons was not necessary for overall rhythmicity, it was sufficient to restore the rhythmicity lost with *Mai179-Gal4*-driven disruption of *per* or *tim*. These results are consistent with recent work suggesting that interactions between clock neurons create multiple independent oscillators that regulate locomotor activity rhythms (*Yao and Shafer, 2014*).

Our results further suggest that the molecular clock needs to be disrupted in both the morning and evening oscillator neurons to disrupt locomotor rhythmicity. When we drove *per^{CRISPR}* or *tim-^{CRISPR}* with *Mai179-Gal4*, which expresses in a subset of clock neurons that include both morning oscillator neurons (s-LNvs) and evening oscillator neurons (primarily 3 CRY⁺ LNds, and the 5th s-LNv), we saw a complete loss of overall rhythmicity. Previous research has shown that rescuing the circadian clock with *UAS-per* expression in a *per* null background in *Mai179-Gal4* cells was not sufficient to fully restore rhythmicity (*Grima et al., 2004*; *Rieger et al., 2009*; *Picot et al., 2007*), but it is possible that UAS-driven expression of *per* did not fully recapitulate endogenous, cyclical expression levels. In contrast, our results demonstrate that the molecular clock in one or more of the *Mai179-Gal4* expressing neurons is necessary for behavioral rhythms. Perhaps the morning and evening oscillators function with some redundancy, coordinating rhythmicity in a distributed, complex network, that only requires a cell-intrinsic molecular clock in one subset of neurons to generate behavioral rhythms. In other words, an intact molecular clock in one subset of clock neurons is able to compensate for loss in another subset, suggesting that clock neurons do not rely entirely on a cell-intrinsic molecular clock to generate behavioral rhythms.

Our results highlight how cell-specific CRISPR-mediated gene disruption can be used to better understand the role the molecular clock plays in specific subsets of circadian neurons to control behavioral rhythmicity. Our work also demonstrates the immense potential of the approach engineered by Port and Bullock to produce cell-specific, CRISPR-mediated gene disruption in somatic cells. These tools provide a new standard for the field and can now be used to investigate the tissue-specific function of circadian genes in both neuronal subsets and 'peripheral clocks' outside the brain that control other circadian-regulated physiologies.

## Materials and methods

**Key resources table**

| Reagent type (species) or resource | Designation | Source or reference | Identifiers | Additional information |
|---|---|---|---|---|
| Genetic reagent (*D. melanogaster*) | *per01* | other | FLYB:FBal0013649 | Obtained from Jaga Giebultowicz |
| Genetic reagent (*D. melanogaster*) | *UAS-sgRNA-acp98AB^{4x}* | this paper | | Available upon request, will be deposited at BDSC |
| Genetic reagent (*D. melanogaster*) | *UAS-sgRNA-per^{4x}* | this paper | | Available upon request, will be deposited at BDSC |
| Genetic reagent (*D. melanogaster*) | *UAS-sgRNA-tim^{3x}* | this paper | | Available upon request, will be deposited at BDSC |
| Genetic reagent (*D. melanogaster*) | *UAS-Cas9.2* | Bloomington Drosophila Stock Center | BDSC:58986 FLYB:FBti0166500 | |
| Genetic reagent (*D. melanogaster*) | *UAS-myr-GFP* | Bloomington Drosophila Stock Center | BDSC:32198 FLYB:FBti0131964 | |
| Genetic reagent (*D. melanogaster*) | *UAS-myr-GFP* | Bloomington Drosophila Stock Center | BDSC:32197 FLYB:FBti0131941 | |
| Genetic reagent (*D. melanogaster*) | *tim-Gal4* | Bloomington Drosophila Stock Center | BDSC:7126 FLYB: FBti0017922 | |

*Continued on next page*

*Continued*

| Reagent type (species) or resource | Designation | Source or reference | Identifiers | Additional information |
|---|---|---|---|---|
| Genetic reagent (*D. melanogaster*) | *repo-Gal4* | Bloomington Drosophila Stock Center | BDSC:7415 FLYB: FBti0018692 | |
| Genetic reagent (*D. melanogaster*) | *Mai179-Gal4* | other | FLYB:FBti0017959 | Obtained from Charlotte Helfrich-Förster |
| Genetic reagent (*D. melanogaster*) | *Pdf-Gal4* | Bloomington Drosophila Stock Center | BDSC:6900 | |
| Genetic reagent (*D. melanogaster*) | *Pdf-Gal80* | other | FLYB:FBtp0019042 | Obtained from Michael Rosbash |
| Recombinant DNA reagent | pCFD6 | Addgene | Cat#73915 | |
| Software, algorithm | Clocklab | Actimetrics | | |
| Software, algorithm | FIJI | PMID: 22743772 | | Open source program |
| Software, algorithm | Prism 8 | GraphPad | | |
| Antibody | Polyclonal Chicken anti-GFP | Abcam | Cat#ab13970 | (1:1000) |
| Antibody | Polyclonal Rabbit anti-Per | PMID: 1613555 | | (1:1000) Obtained from Michael Rosbash |
| Antibody | Polyclonal Rat anti-Tim | PMID: 8625406 | | (1:1000) Obtained from Amtia Sehgal and Michael Young |
| Antibody | Polyclonal Chicken anti-RFP | Rockland Immunochemicals | Cat#600-901-379 | (1:200) |
| Antibody | Monoclonal Mouse anti-PDF | Developmental Studies Hybridoma Bank PMID: 15930393 | Cat#PDF C7 | (1:10) |
| Antibody | Monoclonal Mouse anti-Repo | Developmental Studies Hybridoma Bank PMID: 12167411 | Cat#8D12 anti-Repo | (1:20) |
| Antibody | AlexaFluor 488-conjugated Donkey anti-Chicken | Jackson Immunoresearch | Cat#703-545-155 | (1:200) |
| Antibody | AlexaFluor 594-conjugated Donkey anti-Rabbit | Jackson Immunoresearch | Cat#711-585-152 | (1:200) |
| Antibody | AlexaFluor 647-conjugated Donkey anti-Rat | Jackson Immunoresearch | Cat#712-605-153 | (1:200) |
| Antibody | Cy3-conjugated Donkey anti-Chicken | Jackson Immunoresearch | Cat#703-165-155 | (1:200) |
| Antibody | AlexaFluor 647-conjugated Donkey anti-Mouse | Jackson Immunoresearch | Cat#715-605-151 | (1:200) |
| Sequence-based reagent | *clock-fwd* | This paper | | qPCR primer GGATAAGTCCA CGGTCCTGA |
| Sequence-based reagent | *clock-rev* | This paper | | qPCR primer CTCCAGC ATGAGGTGAGTGT |

*Continued on next page*

Continued

| Reagent type (species) or resource | Designation | Source or reference | Identifiers | Additional information |
|---|---|---|---|---|
| Sequence-based reagent | *period-fwd* | This paper | | qPCR primer CGAGTCCACG GAGTCCACACACAACA |
| Sequence-based reagent | *period-rev* | This paper | | qPCR primer AGGGT CTGCGCCTGCCC |
| Sequence-based reagent | *timeless-fwd* | This paper | | qPCR primer CCGTGGAC GTGATGTACCGCAC |
| Sequence-based reagent | *timeless-rev* | This paper | | qPCR primer CGCAATGGG CATGCGTCTCTG |
| Sequence-based reagent | *Actin5C-fwd* | This paper | | qPCR primer TTGTCTGG GCAAGAGGATCAG |
| Sequence-based reagent | *Actin5C-rev* | This paper | | qPCR primer ACCACTCG CACTTGCACTTTC |

### *Drosophila* strains and maintenance

UAS-sgRNA lines (*w;UAS-sgRNA-tim³ˣ*; *w;UAS-sgRNA-per⁴ˣ*; and *w;UAS-sgRNA-acp98AB⁴ˣ*;) were cloned as described below. The *w;;UAS-Cas9.2* line was obtained from Bloomington *Drosophila* Stock Center (#58986). Two different *UAS-myr-GFP* lines were used (2nd chromosome: Bloomington #32198 and 3rd chromosome: Bloomington #32197). *per⁰¹* nulls were a gift from Jaga Giebultowicz.

Gal4/Gal80 lines: *w;tim-Gal4;* (Bloomington #7126), *w;;repo-Gal4* (Bloomington #7415), *w; Mai179-Gal4;* (Helfrich-Förster Lab), *w;Pdf-Gal4;* (Bloomington #6900), *w;;Pdf-Gal80* (Rosbash Lab). All Gal4 driver lines were outcrossed at least six generations to *w⁻CS* (white-eyed *Canton-S* strain).

All flies were grown and maintained on standard yeast-cornmeal-agar media (Archon Scientific, Glucose recipe: 7.6% w/v glucose, 3.8% w/v yeast, 5.3% w/v cornmeal, w/v 0.6% agar, 0.5% v/v propionic acid, 0.1% w/v methyl paraben, 0.3% v/v ethanol) in a humidity controlled (55–65%) 12:12 Light:Dark incubator at 25°C. Males were collected at 1–3 days old and allowed to mate for 1–2 days before being separated from females. Male flies were 7–11 days old at the start of all behavioral and immunohistochemistry experiments.

### Cloning

Multiple gRNAs targeting *per*, *tim*, or *acp98AB* were constructed as previously described (**Port and Bullock, 2016**). gRNA sequences were selected for predicted target specificity and efficiency according to http://chopchop.cbu.uib.no/ (**Montague et al., 2014**). pCFD6 (Addgene #73915) was digested with BbsI-HF (NEB #R3539S) and gel purified. For each construct, inserts were generated in three separate PCR reactions using *pCFD6* as the template and the primers listed in **Supplementary file 2**. The resulting three inserts and the pCFD6 backbone were then assembled by NEBuilder HiFi DNA Assembly (NEB #E2621L) for each construct. Each construct was integrated at the *Su(Hw)attP5* site (**Pfeiffer et al., 2010**) (Bestgene, Inc) and Sanger sequenced (Genewiz). Sequenced flies revealed a polymorphism in one of the four sgRNA scaffolds in the *UAS-t:sgRNA-tim* flies and thus the line is denoted as *UAS-t:sgRNA-tim³ˣ*.

| Transgene | gRNAs expressed (orientation of target sequence) |
|---|---|
| *UAS-t:sgRNA-per⁴ˣ* | 1. GCTTTTCTACACACACCCGG (5′→3′)<br>2. CACGTGCGATATGATCCCGG (3′→5′)<br>3. GGAGTCCACACACAACACCA (5′→3′)<br>4. TACTCGTCCATAGACCACGC (5′→3′) |
| *UAS-t:sgRNA-tim³ˣ* | 1. *TCTGCTGAAGGAATTCACCG (5′→3′)<br>2. TGTGGCGACCCACATCCGTG (3′→5′)<br>3. GAGAACGCGCTGTACAACTG (3′→5′)<br>4. AAGAGGCCAGCGATATGACG (5′→3′) |

*Continued on next page*

*Continued*

| Transgene | gRNAs expressed (orientation of target sequence) |
|---|---|
| UAS-t:sgRNA-acp98AB$^{4x}$ | 1. GTGTCCCCTTATTCGTGCGG (3'→5')<br>2. CACACTATCAAAGGATGACG (5'→3')<br>3. ATAAGGGGACACACTATCAA (5'→3')<br>4. AGTGTGTCCCCTTATTCGTG (3'→5') |

[*]sgRNA scaffold for gRNA 1 of *timeless* has a one bp deletion (GTTTA... instead of GTTTTA...)

## Circadian locomotor activity

Male flies entrained on a 12:12 LD cycle during development and post-eclosion were placed in individual 5 mm tubes in TriKinetics, Inc *Drosophila* Activity Monitors (DAMs) to record their locomotor activity for two days in 12:12 LD, then for 7–11 days in constant darkness (DD). Activity data from the DD period were summed into 15 min bins using DAM file scan software. Clocklab software (Actimetrics) was used to generate actograms and period measurements. Actograms were blindly scored as rhythmic, weakly rhythmic, or arrhythmic; percentages of each category are reported, except when 'weakly rhythmic' was less than 10% of the population, then it was included with 'rhythmic.' Activity data from the LD day one were summed into 15 min bins and Clocklab software was used to generate average actograms.

- Automated Circadian Analysis: After binning the data exactly as described above, we used Clocklab to generate Lomb-Scargle periodograms with a statistical cutoff of p<0.001. The difference between the amplitude of the peak and the value of the threshold line was calculated and flies were classified as rhythmic if the difference was >150. The % rhythmicity results for all genotypes are reported in *Supplementary file 1*.
- Rhythm Power Analysis: After binning the data as described above, we used Clocklab to generate Chi-square periodograms with a statistical cutoff of p<0.001. The peak height value relative to the threshold line is reported as the 'rhythm power.' Additionally, since an animal cannot display negative rhythmicity, all negative values are represented as '0'. The original values, however, are available in *Supplementary file 1*. A Kruskal-Wallis with a Dunn's multiple comparison post-hoc test was performed to determine statistical differences in rhythm power between *per*- and *tim*-targeted flies and *acp*-targeted controls.
- Anticipation Index (MAI or EAI) Analysis: DAM file scan activity for individual flies was summed into 1 hr bins. An anticipatory index was calculated by dividing the sum of the beam breaks 3 hr immediately preceding 'lights on' (MAI) or 'lights off' (EAI) by the sum of the beam breaks 6 hr preceding 'lights on' (MAI) or 'lights off' (EAI), in LD day 2, DD day 2, and DD days 3–9. All circadian data are representative of at least three biological replicates of at least 8–10 flies each per genotype. A Kruskal-Wallis with a Dunn's multiple comparison post-hoc test was used to compare significance between groups.

Quantification of all circadian locomotor activity data represented in figures is provided in *Supplementary file 4*.

## Immunohistochemistry and confocal microscopy

After 6–9 days of entrainment, flies were decapitated at ZT0 and heads were fixed in 4% paraformaldehyde (Electron Microscopy Sciences #RT15710) in PBS + 0.1% Triton X-100 (PTX) for 40 min at room temperature. Heads were washed in PTX and subsequently incubated on ice. Brains were dissected in PTX and blocked with 4% normal donkey serum (NDS, Jackson ImmunoResearch #017-000-121) in PTX for 90 min at room temperature or overnight at 4°C. After blocking, brains were incubated overnight at 4°C in primary antibody: chicken α-GFP (1:1000, Abcam #ab13970), rabbit α-Per (1:1000, gift of Michael Rosbash [*Liu et al., 1992*]), rat α-Tim (1:1000, gift of Amita Sehgal and Michael Young [*Hunter-Ensor et al., 1996*]), chicken α-RFP (1:200, Rockland 600-901-379), mouse α-repo (1:20, DSHB 8D12), and/or mouse α-PDF (1:10, DSHB C7; *Cyran et al., 2005*) in PTX + 2% NDS. Rabbit α-Per was pre-adsorbed on dechorionated *per$^{01}$* embryos overnight in PTX + 2% NDS prior to use. Brains were washed in PTX and incubated overnight at 4°C in secondary antibody: Alexa Fluor 488–conjugated donkey α-chicken (1:200, Jackson ImmunoResearch #703-545-155), Alexa Fluor 594–conjugated donkey α-rabbit (1:200, Jackson ImmunoResearch #711-585-152), Alexa Fluor 647–conjugated donkey α-rat (1:200, Jackson ImmunoResearch #712-605-153), Cy3–conjugated

donkey α-chicken (1:200, Jackson ImmunoResearch #703-165-155), and/or Alexa Fluor 647–conjugated donkey α-mouse (1:200, Jackson ImmunoResearch #715-605-151) in PTX + 2% NDS. Brains were washed in PTX then PBS and were mounted on coverslips coated with Poly-L-Lysine (0.1 mg/mL, Advanced BioMatrix #5048) and PhotoFlow 200 (0.36%, Kodak #1464510). Coverslips were serially dehydrated with increasing concentrations of ethanol (30, 50, 75, 95, 100, 100%) and cleared with two washes in 100% xylenes. Coverslips were mounted onto slides with DPX (Electron Microscopy Sciences #RT13510) and dried at room temperature overnight before imaging.

Images were acquired on a Zeiss LSM 800 Axio Observer seven inverted confocal microscope (ZEISS) using 488-, 561-, and 647 nm lasers and a Plan-Apochromat 63x/1.40 oil immersion lens (*Figures 3*, *5* and *6*), 40x/1.2 water lens (*Figure 2—figure supplement 5*), or 20x/0.8 dry lens (*Figure 2—figure supplement 4*). Z-stacks were taken using Zeiss LSM confocal software Zen 2.3 (1.5 μm slice thickness except for *Figure 6* where 1.0 μm slice thickness was used). Image analysis was performed in FIJI (*Schindelin et al., 2012*); mean fluorescence intensity of GFP-positive nuclei (*Figures 3* and *5*) or RFP- or PDF-positive nuclei (*Figure 6*) was measured, normalized by subtracting a measurement of mean background intensity, and analyzed using GraphPad Prism software. For *Pdf-Gal4* experiments (*Figure 5*), the number of GFP$^+$ neurons in each brain was counted and analyzed to assess potential CRISPR-driven cytotoxicity. For *Mai179-Gal4; Pdf-Gal80* experiments (*Figure 6*), neurons were visually scored as RFP- and/or PDF-positive for analysis.

Quantification of all IHC data represented in figures is provided in *Supplementary file 4*.

## Quantitative real-time PCR (QRT-PCR)

14 day old male flies previously entrained to 12:12 LD were placed in constant darkness (DD) for 24 hours, after which seven circadian timepoints were taken at CT-1, 5, 9, 13, 17, 21 and 25, snap-frozen in liquid nitrogen, and stored at −80°C. RNA was extracted from 60 heads for each of 4 biological replicates per genotype/timepoint with TRIzol (Invitrogen) following the manufacturer's protocol. Samples were treated with DNaseI (Invitrogen), then heat inactivated. cDNA was synthesized by Revertaid First Strand cDNA Synthesis Kit (Thermo Scientific). PowerUp SYBR Mastermix (Applied Biosystems) was used to perform QRT-PCR using a CFX-Connect thermal cycler (BioRad). Primer efficiency and relative quantification of transcripts were determined using a standard curve of serial diluted cDNA. Transcripts were normalized using Actin5C as a reference gene. The Jonckheere-Terpstra-Kendall (JTK) algorithm was applied using the JTK-Cycle package in R software (*Hughes et al., 2010*) to determine significance of rhythmic cycling. Only *acp*-targeted flies displayed significantly rhythmic cycling of all three genes, indicating oscillation similar to wild-type *tim*, *per*, and *clk* RNA cycling over the circadian day. While *tim-Gal4>tim$^{CRISPR}$* did not result in significant rhythmicity of *per*, *tim*, or *clk* transcript, *tim-Gal4>per$^{CRISPR}$* did result in minor significant rhythmic cycling of the *clk* transcripts, although the peaks are not consistent with those of wild-type animals, indicating arrhythmic expression at the RNA level.

Primer sequences:

*clock*-fwd-GGATAAGTCCACGGTCCTGA
*clock*-rev-CTCCAGCATGAGGTGAGTGT
*period*-fwd-CGAGTCCACGGAGTCCACACACAACA
*period*-rev-AGGGTCTGCGCCTGCCC
*timeless*-fwd-CCGTGGACGTGATGTACCGCAC
*timeless*-rev-CGCAATGGGCATGCGTCTCTG
*Actin5C*-fwd-TTGTCTGGGCAAGAGGATCAG
*Actin5C*-rev- ACCACTCGCACTTGCACTTTC

## Acknowledgements

This work was supported by the following funding sources: NIH R01GM105775 (MSH), R35 GM127049 (MSH), NIH R01GM117407 (JCC), NIH 2T32GM007367-42 (RMO, MSTP training grant), 5T32HL120826 (RD), NIH training grant 5T32DK007328 (MU), and the Charles H Revson Foundation Senior Postdoctoral Fellowship in Biomedical Science (MU).

Stocks obtained from the Bloomington *Drosophila* Stock Center (NIH P40OD018537) were used in this study.

## Additional information

### Funding

| Funder | Grant reference number | Author |
| --- | --- | --- |
| National Institutes of Health | R01GM105775 | Mimi Shirasu-Hiza |
| National Institutes of Health | R35GM127049 | Mimi Shirasu-Hiza |
| National Institutes of Health | R01GM117407 | Julie C Canman |
| National Institutes of Health | 2T32GM007367-42 | Reed M O'Connor |
| National Institutes of Health | 5T32HL120826 | Rebecca Delventhal |
| National Institutes of Health | 5T32DK007328 | Matthew Ulgherait |
| Charles H. Revson Foundation | | Matthew Ulgherait |

The funders had no role in study design, data collection and interpretation, or the decision to submit the work for publication.

### Author contributions

Rebecca Delventhal, Conceptualization, Resources, Data curation, Formal analysis, Supervision, Investigation, Visualization, Methodology, Writing—original draft, Project administration, Writing—review and editing, Conceived of and designed experiments, Performed experiments, Analyzed and interpreted data, Prepared the manuscript; Reed M O'Connor, Conceptualization, Data curation, Formal analysis, Validation, Investigation, Visualization, Methodology, Writing—review and editing, Conceived of and designed experiments, Performed experiments, Analyzed and interpreted data; Meghan M Pantalia, Conceptualization, Data curation, Software, Formal analysis, Validation, Investigation, Visualization, Methodology, Writing—review and editing, Conceived of and designed experiments, Performed experiments, Analyzed and interpreted data; Matthew Ulgherait, Data curation, Formal analysis, Validation, Investigation, Methodology, Writing—review and editing, Performed experiments, Analyzed and interpreted data; Han X Kim, Resources, Methodology, Generated tools and provided technical support; Maylis K Basturk, Resources, Investigation, Performed experiments, Analyzed and interpreted data; Julie C Canman, Visualization, Writing—review and editing, Prepared the manuscript; Mimi Shirasu-Hiza, Conceptualization, Resources, Supervision, Funding acquisition, Writing—original draft, Project administration, Writing—review and editing, Conceived of and designed experiments, Performed experiments, Prepared the manuscript

### Author ORCIDs

Rebecca Delventhal  https://orcid.org/0000-0002-0920-0004
Reed M O'Connor  https://orcid.org/0000-0001-6025-772X
Meghan M Pantalia  https://orcid.org/0000-0001-9932-4926
Julie C Canman  https://orcid.org/0000-0001-8135-2072
Mimi Shirasu-Hiza  https://orcid.org/0000-0002-2730-1765

### Decision letter and Author response

Decision letter https://doi.org/10.7554/eLife.48308.032
Author response https://doi.org/10.7554/eLife.48308.033

## Additional files

### Supplementary files

• Supplementary file 1. Percent rhythmicity values using Lomb-Scargle periodogram automated analysis. Individual flies were classified as rhythmic if difference between amplitude and the value, according to Lomb-Scargle periodogram quantifications, of the best fit line was greater >150, using a p<0.001 statistical cutoff.
DOI: https://doi.org/10.7554/eLife.48308.026

• Supplementary file 2. Primers to clone inserts for *UAS-sgRNA* construct assembly.
DOI: https://doi.org/10.7554/eLife.48308.027

• Supplementary file 3. Summary of all rhythmicity parameters for all experiments.
DOI: https://doi.org/10.7554/eLife.48308.028

• Supplementary file 4. Quantification of all circadian locomotor activity and immunohistochemistry data. Data are grouped into different sheets within spreadsheet by main figure, genotype, and data type (activity or IHC).
DOI: https://doi.org/10.7554/eLife.48308.029

• Transparent reporting form
DOI: https://doi.org/10.7554/eLife.48308.030

### Data availability

All data generated or analyzed during this study are included in the manuscript and supporting files.

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
