## [Decision Letter]

Thank you for submitting your article "Dissection of central clock function in *Drosophila* through cell-specific CRISPR-mediated clock gene disruption" for consideration by *eLife*. Your article has been reviewed by three peer reviewers, and the evaluation has been overseen by a Reviewing Editor and Catherine Dulac as the Senior Editor. The reviewers have opted to remain anonymous.

The reviewers have discussed the reviews with one another and the Reviewing Editor has drafted this decision to help you prepare a revised submission.

Delventhal et al. use targeted CRISPR in *Drosophila* to analyze the role of different subsets of brain clock neurons in rhythmic behavior. It was shown previously, largely through rescue experiments, that expression of the PER clock protein in only PDF cells (LNvs) could drive morning activity in light:dark cycles (LD) as well as sustained rhythms in constant darkness (DD). On the other hand, PER expression in LNd neurons + the 5th small LNv drives locomotor activity in the evening. This study confirms the specific ability of PDF cells to drive the morning peak in LD and of E cells to drive evening activity, but interestingly shows that a clock in the PDF cells is actually not required for behavioral rhythms in DD. It appears that clocks in different neuronal clusters can act redundantly with respect to sustaining rhythms in constant darkness.

This is a nice study that challenges the model of morning cells being critical for free running rhythms in DD. The authors develop efficient CRISPR tools to do these experiments and the results are convincing. The findings are important and will be of significant interest to the field. The data are clearly presented and the manuscript is well-written.

Essential revisions:

The reviewers requested more detail about the behavioural phenotypes, specifically effects of manipulating each neuronal cluster on different parameters of rhythms (rhythm strength, periodicity,% rhythmicity). The authors should also show this for the CRISPR tools on their own (e.g. CAS9 over-expression) as these tools will likely be used by circadian researchers in the future.

Along the same lines, reviewers felt that GFP positive cell data for PER and TIM expression in the Mai-179 mediated knockout experiment should not have been pooled. It would be much more informative to treat the s-LNvs, 5th s-LNv and the LNds separately. One might imagine that GAL4 expression strength might vary from class to class. Are the outliers in E and F predominantly from one class? Also, the immunohistochemistry data in Figure 3—figure supplement 3 show substantial expression of PER and TIM in the *tim*-crispr mutants in both LNds and LNvs. Is this because the outliers are shown? It is important to clarify this.

The authors claim that loss of PER in glia does not affect rhythms. However, they need to provide evidence of successful knockdown in glia. Even though this was done by the Jackson laboratory, it should be demonstrated in the context of these experiments. Also, it should be noted that not all glia are *repo* positive, so a role for glia cannot be entirely excluded.

---

## [Author Response]

Essential revisions:The reviewers requested more detail about the behavioural phenotypes, specifically effects of manipulating each neuronal cluster on different parameters of rhythms (rhythm strength, periodicity,% rhythmicity). The authors should also show this for the CRISPR tools on their own (e.g. CAS9 over-expression) as these tools will likely be used by circadian researchers in the future.

We thank the reviewer for their thoughtful critique and helpful suggestions. We added new behavioral analysis of the parental controls (see Figure 2—figure supplement 3) to address their concerns. We also reanalyzed existing datasets to measure rhythm strength as the height of the chi-square peak (see figure supplements 1 for Figures 2, 3, 4, and 6). The% rhythmicity and periodicity is reported in each figure displaying rhythmicity data for all genotypes tested. This information is also available in Supplementary file 3.

Along the same lines, reviewers felt that GFP positive cell data for PER and TIM expression in the Mai-179 mediated knockout experiment should not have been pooled. It would be much more informative to treat the s-LNvs, 5th s-LNv and the LNds separately. One might imagine that GAL4 expression strength might vary from class to class. Are the outliers in E and F predominantly from one class? Also, the immunohistochemistry data in Figure 3—figure supplement 3 show substantial expression of PER and TIM in the tim-crispr mutants in both LNds and LNvs. Is this because the outliers are shown? It is important to clarify this.

In response to this comment, we reanalyzed this dataset by grouping LNds, s-LNvs, and l-LNvs separately. Though we could not reliably distinguish the 5^th^ s-LNv from the other s-LNvs, we found that in any case the majority of any residual Per/Tim signal was located in the l-LNvs (see Figure 3—figure supplement 4). This is consistent with previous reports that the l-LNvs have weak and variable expression of *Mai179-Gal4* (Siegmund and Korge, 2001, Yao et al., 2016). This relates to the reviewers’ second question on the expression of Per and Tim in *tim*-CRISPR mutants (Figure 3—figure supplement 3). Some LNds and l-LNvs continue to express Per and Tim because these cells do not substantially express *Mai179-Gal4* (or the UAS-driven CRISPR constructs), as evidenced by the absence of GFP expression (there is variable expression in the l-LNvs and LNds). The loss of circadian locomotor activity due to *Mai179-Gal4* driven CRISPR is thus due to loss of Per and Tim in targeted cells. The figure has now been updated to clarify the distinction between targeted and non-targeted cells in each image shown. In the quantification, only nuclei of GFP-positive neurons were measured.

The authors claim that loss of PER in glia does not affect rhythms. However, they need to provide evidence of successful knockdown in glia. Even though this was done by the Jackson laboratory, it should be demonstrated in the context of these experiments. Also, it should be noted that not all glia are repo positive, so a role for glia cannot be entirely excluded.

We have now validated the efficiency and specificity of the CRISPR deletion in Repo^+^ glial cells (see Figure 2—figure supplement 5). We also changed the text to accurately reflect our targeted cell population, removing the claim that *repo-Gal4* is a “pan-glial” driver. The text now reads: “…we CRISPR-targeted *tim* and *per* in Repo^+^ glia, using the glial driver *repo-Gal4*.”